# A Long-Term Video Tracking Method for Group-Housed Pigs

**DOI:** 10.3390/ani14101505

**Published:** 2024-05-19

**Authors:** Qiumei Yang, Xiangyang Hui, Yigui Huang, Miaobin Chen, Senpeng Huang, Deqin Xiao

**Affiliations:** 1College of Mathematics and Informatics, South China Agricultural University, Guangzhou 510642, China; yqmbegonia@scau.edu.cn (Q.Y.); scau_hxy@stu.scau.edu.cn (X.H.); hyg2021scau@stu.scau.edu.cn (Y.H.); 20213162019@stu.scau.edu.cn (M.C.); hsp2454@stu.scau.edu.cn (S.H.); 2Key Laboratory of Smart Agricultural Technology in Tropical South China, Ministry of Agriculture and Rural Affairs, Guangzhou 510642, China

**Keywords:** pig, object detection, deep learning, multi-object tracking

## Abstract

**Simple Summary:**

Simple Summary: Pig tracking provides strong support for refined management in pig farms and can assist in realizing customized, automated, and intellectualized management. However, there are many complex factors in actual production that result in pigs not being tracked consistently over time. To address this issue, we proposed a long-term video tracking method for group-housed pigs based on improved StrongSORT. The experimental results proved that our study demonstrates high practicality, catering to the needs of practical production. It improved the efficiency of farming personnel and lays the foundation for the automated monitoring of pigs. Additionally, it is worth mentioning that we have also released a dataset specifically designed for long-term pig tracking in this paper.

**Abstract:**

Pig tracking provides strong support for refined management in pig farms. However, long and continuous multi-pig tracking is still extremely challenging due to occlusion, distortion, and motion blurring in real farming scenarios. This study proposes a long-term video tracking method for group-housed pigs based on improved StrongSORT, which can significantly improve the performance of pig tracking in production scenarios. In addition, this research constructs a 24 h pig tracking video dataset, providing a basis for exploring the effectiveness of long-term tracking algorithms. For object detection, a lightweight pig detection network, YOLO v7-tiny_Pig, improved based on YOLO v7-tiny, is proposed to reduce model parameters and improve detection speed. To address the target association problem, the trajectory management method of StrongSORT is optimized according to the characteristics of the pig tracking task to reduce the tracking identity (ID) switching and improve the stability of the algorithm. The experimental results show that YOLO v7-tiny_Pig ensures detection applicability while reducing parameters by 36.7% compared to YOLO v7-tiny and achieving an average video detection speed of 435 frames per second. In terms of pig tracking, Higher-Order Tracking Accuracy (HOTA), Multi-Object Tracking Accuracy (MOTP), and Identification F1 (IDF1) scores reach 83.16%, 97.6%, and 91.42%, respectively. Compared with the original StrongSORT algorithm, HOTA and IDF1 are improved by 6.19% and 10.89%, respectively, and Identity Switch (IDSW) is reduced by 69%. Our algorithm can achieve the continuous tracking of pigs in real scenarios for up to 24 h. This method provides technical support for non-contact pig automatic monitoring.

## 1. Introduction

Pig tracking is of significant importance in pig farm management. Through the individual localization and tracking of pigs, it is possible to collect data on pig movements, analyze pig behavior, and monitor various pig-related information, to find the abnormal pigs in a timely manner [1,2]. At the same time, the automatic tracking and monitoring of pigs can avoid labor-intensive and error-prone manual management, thereby improving management efficiency [3,4,5].

Current research for pig tracking mainly includes contact methods and non-contact methods. The contact methods realize tracking using sensor devices such as ear tags worn by pigs. Eisermann et al. [6] used an UHF-RFID system to individually identify pigs and combined a single point load cell (SPLC) to assess the individual manipulative behaviour of group-housed pigs. Knoll et al. [7] mounted UWB tags on pig ear tags and worked with UWB tags in each pen to achieve the accurate positioning of individual pigs. Lee et al. [8] developed a monitoring system to estimate the location and movement of each pig using Bluetooth low-energy (BLE) tags and wireless broadband leaky coaxial cable (WBLCX) antennas. However, the sensors used in these methods are costly, easily damaged, and detached. In addition, the pigs tend to have stress reactions during sensor installation, which in turn affects their welfare.

In contrast, using non-contact tracking methods based on video imagery can help avoid these issues. The current research on image-based pig tracking algorithms is mainly divided into manual feature-based tracking methods and deep learning-based tracking methods. For manual feature-based tracking methods, Xiao et al. [9] identified pigs based on color information, removed the noise through the connected region of the binary image, and proposed a set of association rules with constraints (DT-ACR) for pig tracking. Sun and Li [10] designed a multi-pig tracking algorithm based on joint probability data association and particle filtering. Although such methods can track pigs, they need to define features manually, which have poor generalization, and the tracking performance is easily degraded in real-world scenarios due to challenges such as insufficient light and pigs sticking to and obscuring each other [11]. These limitations make it difficult to meet the needs of the actual scene.

Deep learning-based methods have the advantage of automatically extracting and learning features, which allows models to exhibit strong robustness even under various challenging conditions such as varying lighting and occlusion [12]. Li et al. [13] proposed a correlation matrix combining the embedding cosine distance and GIoU distance to address tracking failures caused by occlusion or temporary target loss. Tu et al. [14] presented a multi-object tracking method that simultaneously outputs classification, bounding box, and appearance information to complete the behavioral tracking of group-housed pigs. The above methods are based on the Joint Detection and Embedding (JDE) paradigm. Although they have good real-time tracking performance, their tracking accuracy is generally not as good as the Tracking-by-Detection (TBD) method.

The principle of the TBD method is to treat object detection and object tracking as two separate steps, and employ a correlation matching algorithm to associate the detected object candidates with existing tracked targets [15]. Van der Zande et al. [16] used Yolo v3 to detect pigs and then tracked them by the SORT algorithm. Zhang et al. [17] combined the CenterNet detection algorithm with an improved DeepSORT algorithm to track weaned piglets. Gong et al. [18] proposed an improved IOU-tracker combined with YOLOv5s for real-time pig tracking. Shirke et al. [19] introduced a multi-camera pig tracking system that initially employed YOLOv4 for pig detection, followed by using homography estimation to locate the same pig in different camera regions, and finally tracked pigs using the DeepSORT algorithm. These methods are prone to switching pig identities due to prolonged occlusions, making it challenging to achieve long-term continuous tracking.

To address the above issues, we propose a multi-pig long-term tracking algorithm based on improved StrongSORT, which aims to achieve the long-term and continuous tracking of individual pigs in real-world scenarios. Furthermore, this study explored the significance of pig tracking in the field of pig farming to satisfy actual production demands.

## 2. Materials and Methods

### 2.1. Video Acquisition

Both self-collected videos and the public videos [20] were used for our research. The self-collected videos were captured from two pig farms in Luogang and Xinxing, Guangdong, China.

The pen selected from Luogang Pig Farm has 4 pigs, and the pen size is 4 m × 5 m. Landrace gilts were selected as experimental subjects, and they were continuously monitored 24 h a day. The video was taken from July 2016 to August 2016. The pen selected from Xinxing Pig Farm has 7 pigs, and the pen size is 2 m × 4 m. The experimental videos were captured between July 2020 and August 2020. The public videos include pens with 7 to 16 pigs each, including a total of 15 video clips during the daytime and nighttime, each with a length of 30 min. The information of each video set is shown in Table 1.

The video data were captured from the top view and covered the entire pen, as shown in Figure 1. There were no pigs disappearing within the camera’s field of view. In order to facilitate the manual identification of different pigs in subsequent experiments, the pigs in Luogang Pig Farm were marked on their backs with “A”, “B”, “C”, and “D”, as shown in Figure 1a. Similarly, the pigs in Xinxing Pig Farm were marked on their backs with labels “1”, “2”, “3”, “4”, “5”, “6”, and “7”, as shown in Figure 1b.

### 2.2. Data Preprocessing and Dataset Construction

This study constructed datasets for pig detection and multi-pig tracking. The former was used to train the pig detection model, while the latter was employed to evaluate multi-pig tracking algorithms.

#### 2.2.1. Data Preprocessing

Due to the camera’s angle of view, the acquired videos contained corridors or other pens where unrelated pigs may be detected during the pig detection process. It can seriously affect the experimental results. To overcome this, a mask operation was performed on videos with these issues (Luogang Pig Farm videos and public videos). The specific operation is illustrated in Figure 2, where the green box area is the main monitoring area and the red box area signifies areas that can potentially affect the experimental results. After preprocessing, the red box area is covered by a black mask while the green area is retained. Figure 2a shows the mask for Luogang Pig Farm, which mainly covers pigs moving in the corridor and pigs in other pens. And Figure 2b shows the mask for the public videos, primarily covering pigs in neighboring pens.

#### 2.2.2. Pig Detection Datasets

Our public video dataset was taken from reference [20], which can be downloaded at http://psrg.unl.edu/Projects/Details/12-Animal-Tracking (accessed on 10 December 2022). To create the pig detection dataset, we extracted image frames from the captured videos. For each pig farm, we selected 1800 images with high clarity and low similarity and included four different environments: daytime crowded, daytime sparse, nighttime crowded, and nighttime sparse. These images were manually annotated using LabelImg software(v1.8.6). Finally, the annotated images were randomly divided into a training set and testing set following a ratio of 8:2, as detailed in Table 2.

#### 2.2.3. Pig Tracking Datasets

For pig tracking, we selected 10 video clips of group-housed pigs, as detailed in Table 3. The selected videos included different lighting conditions during the day and night, covering three different pig farm scenarios, with low and high frame rates. The density of pigs within the pens varied as well. These videos ranged in length from 2 min to 24 h, and notably, the 24 h pig tracking video was the longest in the literature currently available in the field. The selected videos were annotated frame by frame using DarkLabel software(v 2.4), creating a bounding box for each pig and ensuring the same identification number (ID) was assigned to the same pig across adjacent frames.

### 2.3. Multi-Pig Long-Term Tracking Methods

#### 2.3.1. Framework of Multi-Pig Long-Term Tracking Method

In practical pig farm management, there is a need for highly accurate and stable pig tracking algorithms. The TBD paradigm can meet this demand as it is capable of providing highly accurate tracking results. Our study used the TBD paradigm.

The multi-pig tracking algorithm proposed in this paper was divided into two modules: pig detection and multi-pig tracking. Its framework is shown in Figure 3. Firstly, the video sequence was input into the pig detector YOLO v7-tiny_Pig to detect and obtain pig location. Then, the detection result was fed into the optimized StrongSORT [21] multi-object tracker for feature extraction to obtain the pig appearance feature. Simultaneously, NSA Adaptive Kalman filtering was employed to predict and update pig trajectories. Finally, the Hungarian algorithm was utilized to globally match the pig appearance feature, pig location, and pig trajectories, thus accomplishing multi-pig tracking.

#### 2.3.2. Lightweight Pig Detector YOLO v7-Tiny_Pig

Since our tracking algorithm was based on the TBD paradigm, the performance of the pig detector directly impacts the effectiveness of tracking. Currently, the YOLO series detectors have shown excellent performance. Therefore, we used the most popular YOLO v7 [22] series network for pig detection. It is a one-stage object detection algorithm and offers three basic models designed for different scenarios: YOLO v7-tiny is suitable for edge GPUs, YOLO v7 is designed for regular GPUs, and YOLO v7-W6 is intended for cloud GPUs. The accuracy of these three basic models increases successively, while the number of parameters increases.

On the one hand, due to the fact that the object detection in this study only involves one category, which is simple, there was no need for complex network models. On the other hand, for further analysis of the pig tracking results, a faster processing speed was required. Therefore, the YOLO v7-W6 model was excluded. To further select an appropriate pig detection model, we used the aforementioned pig detection dataset to train and test the other two models YOLO v7 and YOLO v7-tiny, respectively. After experimental comparison, the lighter YOLO v7-tiny model was chosen as the base model, with the network structure shown in Figure 4a.

The YOLO v7-tiny network consists of four parts, including the Input, Backbone, Neck, and Output. The purpose of the Input stage is to feed the data into the Backbone after performing data augmentation. The Backbone consists of convolutional layers and the C5 structure, which feeds the extracted features into the Neck for feature fusion. The network structure of C5 is illustrated in Figure 4b. The Neck consists of the CBC structure (Figure 4c), C5, and convolutional layers, and it performs downsampling to reduce feature dimensionality. The Output includes three feature maps of different scales predicting targets of different sizes, and finally feature fusion is performed to output the detection results.

Since the cameras in our experiment were installed directly above the center of the pig pens, and the proportion of pigs in any location within the pen relative to the entire image did not vary significantly, a single scale YOLOHead can complete the task of detecting pigs [23]. Therefore, we made modifications to YOLO v7-tiny by removing the shallow and deep feature maps, retaining only the mid-level feature maps to obtain the pig detection network YOLO v7-tiny_Pig. The improved network structure reduced the number of parameters and accelerated the detection speed. The network structure is illustrated in Figure 5.

#### 2.3.3. Multi-Pig Tracker StrongSORT-Pig

The multi-pig tracker StrongSORT-Pig proposed in this study is an improved version of the StrongSORT algorithm, tailored to the specific characteristics of the real scenario. The StrongSORT algorithm is an improvement on the DeepSORT [24] algorithm, which is further developed based on the SORT [25] algorithm. The SORT algorithm uses only motion information for multi-object tracking, which can lead to ID switches when targets are occluded. The DeepSORT algorithm introduces a re-identification model, combines motion and appearance information, and employs a cascade matching strategy, thereby improving matching accuracy. The StrongSORT algorithm optimizes the motion branch and the appearance branch of the DeepSORT algorithm separately, and introduces advanced methods and strategies.

Since the pig pens in our experiment were enclosed spaces with restricted access and the number of pigs did not change over time, this study improved the StrongSORT algorithm and optimized the trajectory management strategy. These improvements aimed to reduce the number of ID switches, allowing pigs to be retracked even after prolonged occlusion, thus extending the tracking time. Figure 6 shows the algorithmic process of StrongSORT-Pig.

The StrongSORT-Pig ensures that the same pig maintains the same ID throughout the time sequence by globally matching the pig detection results with predicted trajectories. The matching result is jointly determined by both the motion branch and the appearance branch, represented by the blue and yellow arrows in Figure 6, respectively.

The motion branch

Due to the random and irregular movement behaviors of pigs, it can easily lead to image blurring and thus affect the detection and tracking performance. To address this issue, the motion branch in this study was divided into two stages. In the first stage, we adopted the Enhanced Correlation Coefficient (ECC) model to compensate the blurred images caused by pig movements. In the second stage, the NSA Kalman filtering algorithm was applied to predict and smooth the motion trajectories of each pig through dynamically adjusting the noise covariance based on detection confidence scores. It consisted of two phases: the prediction phase and the update phase. In the prediction phase, the current pig trajectory state was predicted based on the pig trajectory state in the previous frame and the noise matrix in the current frame. In the update phase, the trajectory state was updated using the results from the prediction stage, along with the noise of the pig detection and the pig detection results in the current frame. The confidence of pig detection results affected the accuracy of state updates. It is worth noting that Kalman filtering was only used to predict the pig’s trajectory before the matching operation. The entire process made full use of the pig motion state from the previous frame and the current pig detection results. By dynamically and adaptively adjusting the weights of the detection results, the impact of various noises on the tracking results was effectively reduced, thereby making the pig tracking results more accurate.

2.The appearance branch

The appearance features of pigs, such as body color and body shape, are relatively stable. However, in a video stream, the appearance of the same pig in different frames may change slightly due to factors like lighting changes. Additionally, due to the small differences between individual pigs, it is necessary to establish a sensitive pig appearance feature extractor to capture highly discriminate features, thereby improving the performance of the tracker. On the other hand, the Exponential Moving Average (EMA) feature update strategy can update the trajectory appearance of the corresponding pigs based on the detailed features obtained by the feature extractor, which helps to smoothly adapt the appearance changes and improve the accuracy and robustness of tracking. Therefore, the appearance branch contained the pig appearance feature extractor and the feature update strategy based on the EMA. Equation (1) represents the appearance state $e_{i}^{t}$ of the i-th trajectory updated based on the EMA strategy in the t-th frame:(1)$$e_{i}^{t}=\alpha e_{i}^{t-1}+\left(1-\alpha \right)f_{i}^{t}$$
where $e_{i}^{t-1}$ is the appearance state of the ith trajectory in the (t−1)th frame, $f_{i}^{t}$ is the appearance feature matched to the ith trajectory (i.e., the current trajectory) in the tth frame (i.e., the current frame), and α is the weight parameter.

The feature update strategy was able to efficiently update the appearance state of the trajectory by combining the appearance features detected in the current frame while preserving the appearance information of the previous video frames. It can reduce the impact of detection noise and rapid changes in the tracked targets on appearance information and capture changes in appearance features between video frames, thereby improving the performance of the tracking system.

3.The global matching process

During the global matching stage, different weights were assigned to appearance and motion information to calculate the pairwise association costs between all pigs and trajectories, thereby solving the assignment problem between targets and trajectories.

For the appearance information, the minimum cosine distance $d_{\alpha }$ was used as the matching cost, and it was computed between the current detection $r_{j}$ and the appearance of all trajectories, following the rule defined in Equation (2).
(2)$$d_{\alpha }=min\{1-r_{j}^{T}e_{i}^{t-1}|i\in R\}$$
where $r_{j}$ is the appearance state of the jth detection, $e_{i}^{t-1}$ is the appearance state of the ith trajectory in the (t−1)th frame, and R is the number of all trajectories.

For the motion information, the Mahalanobis distance $d_{m}^{M}$ between the center point of the pig detection bounding box and the center point of the pig track bounding box was used as a gate to filter out unlikely matches. Then, the cosine distance $d_{m}$ of motion information was used as the matching cost for further matching. The final cost matrix was the weighted sum of appearance and motion costs, which was calculated as shown in Equation (3).
(3)$$C=\lambda A_{\alpha }+(1-\lambda )A_{m}$$
where $A_{\alpha }$ represents the appearance cost matrix, which is the set of $d_{\alpha }$. Similarly, $A_{m}$ represents the motion cost matrix, which is the set of $d_{m}$, and λ is the weight factor.

When pigs are obstructed by other objects or other pigs, the original trajectories and the detection results cannot be matched. This leads to the generation of more trajectories, causing a continuous increase in pig identity IDs. The actual maximum ID value is far greater than the actual number of pigs in the pigpen, resulting in tracking failures. To address this issue, this study optimizes the trajectory deletion and generation process, taking into account the particular scenario of pigsties.

Since the number of pigs in the pen does not suddenly decrease, the maximum matching lifetime is not set for unmatched tracks in our research. Unmatched trajectories will never be deleted to reduce ID switching caused by long-term occlusion. Similarly, as the number of pigs in the pigpen does not suddenly increase, we introduce additional constraints on the generation of new trajectories. The total number of pigs in the pigpen is set as a threshold for the maximum value of ID increase. New trajectories can be generated only when the number of trajectories is less than the threshold, which can limit the endless increase in ID.

By combining the above two optimizations, on the one hand, trajectory deletion is prohibited, and on the other hand, ID growth is restricted. This approach is applied to multi-pig tracking. The specific matching process is shown in Figure 7.

(a)The matching of the motion branch is performed by calculating the Mahalanobis distance between the center point of the pig detection bounding box and the center points of all pig trajectory bounding boxes. If the distance is less than the threshold, the combined matching of motion information and appearance information is performed. If the distance between the pig detection box and all the trajectories is greater than the threshold, the detection box is in the unmatched state, and it is further judged whether it meets the conditions for generating new trajectories. The number of available tracks is less than the total number of pigs.(b)An integrated matching of motion and appearance information is performed, a combined cost matrix is constructed for both, and then the Hungarian algorithm is used for global matching. If the match is successful, the process continues to the next step. If there is no suitable match, the detection box is in the unmatched state, and further judgment is made to determine whether it meets the conditions for new trajectory generation. If it meets the conditions, a new track is generated; otherwise, the threshold is increased, and the matching process is repeated.(c)For successfully matched detection boxes and trajectories, the motion and appearance models of the corresponding trajectories are updated. For unmatched detection boxes, new trajectories are initialized.(d)The next frame of the image is proceeded to and the above process is repeated.

4.Correction algorithm of multi-pig tracking performance

Due to adhesion and occlusion between pigs, some pigs may not be detected at certain times. It can cause pig trajectories to be lost and their location to not be recorded during the missing period. However, when such pigs are detected again, their trajectories reappear. Associating the trajectories before and after missing and filling in the missing detections are crucial for the tracking results. Therefore, two correction algorithms, the appearance-free link model (AFLink) and Gaussian-smoothed interpolation (GSI), which are based on motion information and are independent of appearance information, are incorporated in the tracking procedure.

AFLink utilizes spatiotemporal information to predict the correlation between two trajectories as a way to solve the problem of correlating the trajectories before and after missing. As shown in Figure 8, it takes two trajectories $T_{i}$ and $T_{j}$ as inputs and is calculated by Equation (4).
(4)$$T_{*}={\left\{f_{k},x_{k},y_{k}\right\}}_{k=1}^{N}$$
where $x_{k}$ and $y_{k}$ represent the position coordinates of the nearest N=30 frames of $f_{k}$, and if N is less than 30, it is padded with zeros. The temporal module extracts feature information along the temporal dimension. Then, the fusion module integrates feature information from different dimensions and compresses the information of two trajectories into feature vectors, respectively. Finally, these feature vectors are concatenated to predict the confidence score of the association by a Multilayer Perceptron (MLP), to determine whether the two trajectories belong to the same track. In the association process, spatiotemporal constraints are used to filter out unreasonable trajectory pairs. Then, the global association is decomposed into a linear assignment task using the predicted confidence scores for associations.

GSI employs Gaussian process regression to simulate nonlinear motion and fits undetected trajectory updates using Gaussian-smoothed interpolation, given the tracking trajectories before and after the lack of detection, assuming that the actual motion follows a Gaussian process, based on the properties of Gaussian processes, fitting its Gaussian process distribution based on the given observed trajectories, and using this information to predict and fill in the missing trajectories at the time of detection loss.

The application of AFLink and GSI can significantly improve the robustness of pig tracking algorithms in complex environments. It substantially reduces the loss of pig tracking targets caused by occlusions and unstable detections, resulting in higher tracking accuracy. The stability and continuity of the pig tracking algorithm have also been enhanced.

## 3. Results and Analysis

### 3.1. Experimental Platform and Parameter Settings

The experiments were conducted on the Windows 10 platform, equipped with a 2.50 GHz 12th-Gen Intel(R) Core (TM) i5-12400F processor, 16 GB of memory, and a 1 TB hard drive. The GPU used was an NVIDIA GeForce RTX 3060. The programming language used was Python 3.8, and the deep learning framework used was PyTorch 1.11.0.

### 3.2. Evaluation and Analysis of Pig Detection

#### 3.2.1. Pig Detection Evaluation Metrics

In the pig detection stage, the performance of the pig detection model was evaluated using five metrics: Precision (*P*), Recall (*R*), *F*1 Score, FPS, and *mAP*. The mathematical expression of these metrics is given by Equations (5)–(9).
(5)$$P=\frac{TP}{TP+FP}\times 100\%$$
(6)$$R=\frac{TP}{TP+FN}\times 100\%$$
(7)$$F1=\frac{2PR}{P+R}\times 100\%$$
(8)$$FPS=\frac{frameNum}{elapsedTime}$$
(9)$$mAP=\int_{0}^{1}P(R)dR$$
where TP (True Positives) refers to the number of pigs correctly detected; FN (False Negatives) represents the number of pigs that were not detected; FP (False Positives) is the number of pigs incorrectly detected; frameNum is the total number of frames; and elapsedTime represents the total processing time. This study used single target detection, so calculating mAP means calculating the area under the P-R curve.

#### 3.2.2. Comparison Experiments of Different Models

During the training procedure, we employed the pre-trained YOLO v7-tiny model. The input image size was set to 1280 pixels × 1280 pixels, with a batch size of 8, a learning rate of 0.01, momentum of 0.937, and 150 epochs. Adam optimizer was used to iteratively optimize network parameters.

In order to select a more suitable pig detection model, this study used the same pig detection dataset to conduct experiments in YOLO v7, YOLO v7-tiny, and the YOLO v7-tiny_Pig network, respectively. The experimental results are shown in Table 4.

From Table 4, it can be observed that although the accuracy of YOLO v7-tiny_Pig is slightly lower than those of YOLO v7 and YOLO v7-tiny, it still exceeds 99%, which can meet the requirements of pig tracking. Meanwhile, it has a higher FPS, which processes images 3.4 times faster than YOLO v7, and the number of model parameters is about 10% of YOLO v7. Compared to YOLO v7-tiny, YOLO v7-tiny_Pig processes 65 more frames per second and reduces the number of model parameters by 36.7%. To meet the requirements of real-world production, the pig tracking algorithm needs to have a shorter processing time and can be deployed on edge devices, which requires higher detection speed and smaller model size. Therefore, YOLO v7-tiny_Pig is more suitable for pig detection in this study.

#### 3.2.3. Analysis of Pig Detection Using YOLO v7-Tiny_Pig

In this section, the performance of the proposed pig detection method is further validated through manual observation.

The detection results of pigs in four different environments using YOLO v7-tiny_Pig are shown in Figure 9. Figure 9a,c show the detection results for the sparse pig scenarios during the day and night. The model is able to accurately detect the pig positions, even in Figure 9c where the pigs have a similar appearance to the floor color at night. Figure 9b shows the detection results for the dense pig scenario during the daytime, where pigs are heavily occluded, making it difficult for human observers to distinguish them. However, the model is still able to accurately and quickly detect the pigs. In Figure 9d, it displays the detection results for the dense pig scenario at night. Not only can the pigs in the light part be detected accurately, but the pigs in dark areas are also not missed by the model. Overall, YOLO v7-tiny_Pig demonstrates the ability to accurately detect pigs in different environments, laying a foundation for subsequent multi-pig tracking in natural scenarios.

### 3.3. Evaluation and Analysis of Multi-Pig Tracking Results

#### 3.3.1. Multi-Pig Tracking Evaluation Metrics

HOTA [26], MOTA [27], IDF1 [28], MOTP, and IDSW were chosen to evaluate the multi-pig tracking model.

HOTA considers the accuracy of target matching, tracking, and localization simultaneously, which can evaluate the performance of the tracker more comprehensively and accurately. Particularly in cases of target overlap and occlusion, it is more consistent with the human visual assessment of tracking performance. Its mathematical expression is given by Equation (10).
(10)$$HOTA=\sqrt{\frac{\sum_{c\in \{TP\}}A_{\left(c\right)}}{\left|TP\right|+\left|FN\right|+\left|FP\right|}}$$
(11)$$A_{\left(c\right)}=\frac{\left|TPA_{\left(c\right)}\right|}{\left|TPA_{\left(c\right)}\right|+\left|FNA_{\left(c\right)}\right|+\left|FPA_{\left(c\right)}\right|}$$
where $A_{\left(c\right)}$ is the association accuracy, defined by Equation (11). $TPA_{\left(c\right)}$ is the accuracy of correct correlation, $FNA_{\left(c\right)}$ represents the accuracy of targets that are related but not successfully matched, and $FPA_{\left(c\right)}$ represents the accuracy of completely unrelated false detections.

The mathematical expression of *MOTA* is given by Equation (12), where IDSW represents the number of times the target label ID switched during tracking in frame *t*; gt represents the number of targets observed in frame *t*.
(12)$$MOTA=1-\frac{\sum_{t}\left(FN+FP+IDSW\right)}{\sum_{t}g_{t}}$$
where IDF1 represents the proportion of detected targets that acquire the correct IDs. It is used to measure the overall performance of target identification and tracking, and its mathematical expression is given by Equation (13).
(13)$$IDF1=\frac{2IDTP}{2IDTP+IDFP+IDFN}$$
where IDTP represents the total number of pigs tracked correctly with unchanged IDs, IDFP represents the total number of pigs tracked incorrectly with unchanged IDs, and IDFN represents the total number of pigs tracked missing with unchanged IDs.

MOTP is used to measure the accuracy of the tracker in terms of target position estimation and its mathematical expression is given by Equation (14), where $d_{i,t}$ represents the distance between the pig detection box and the hypothetical pig bounding box in frame *t*; $c_{t}$ represents the total number of matches between pig and hypothesis positions in frame t, and i represents the current detection target.
(14)$$MOTP=\frac{\sum_{t,i}d_{i,t}}{\sum_{t}c_{t}}$$

#### 3.3.2. Comparison of Effects before and after Improvement in the Algorithm

To better demonstrate the performance of the improved tracking algorithm in this study, nine videos (Video1–Video9) from the pig tracking dataset were used to compare the performance of the original StrongSORT, the optimized StrongSORT-Pig tailored for pig farming characteristics, and StrongSORT-Pig_Plus with the inclusion of correction algorithms. The model performance test results are shown in Table 5.

Compared with the original StrongSORT, the StrongSORT-Pig algorithm had shown varying degrees of improvement in most metrics, especially its average value of IDF1 that reached 91.1%, which was improved by 10.57%. The IDSW had reduced by 68 times, which was a 39.78% reduction compared with the original algorithm. In addition, HOTA was improved by an average of 5.71%, indicating the overall superior performance of our method. By incorporating the correction module, the performance of our method was further improved. HOTA was improved by 6.19% compared with the original algorithm. IDF1 was improved by 10.89%, and IDSW was reduced by 118 times, which is a decrease of 69%. This indicated that the correction module significantly enhanced the stability of pig tracking through the predictive interpolation of missing frames.

There was no significant improvement in the MOTA and MOTP metrics, and in some cases, there was a slight decrease, but they still reached 97.6% (MOTA) and 90.88% (MOTP), respectively. This ensured that the algorithm can correctly detect and track the majority of targets with relatively few false positives and false negatives. It can also estimate the position of the target accurately.

In addition, among the three algorithms, StrongSORT-pig had the highest FPS, while StrongSORT-Pig_Plus had the lowest FPS, which does not meet the speed needs of realistic scenarios. This is because the correction algorithm used in the StrongSORT-Pig_Plus model is more time-consuming, resulting in a lower FPS. Therefore, taking both accuracy and FPS into account, we used the StrongSORT-Pig algorithm in the subsequent experiments.

The tracking results of StrongSORT-pig are shown in Figure 10. The 77th, 110th, and 190th frames from Video1 were selected for comparison before and after optimization. In Figure 10a, both pig 7 and pig 15 can be detected in the 77th frame, but they cannot be detected in the 110th frame. They reappeared in the 190th frame, but their IDs had changed. Pig 7 was now identified as pig 18, and pig 15 was identified as pig 17, exceeding the total number of pigs, which was 16. In Figure 10b, both pig 7 and pig15 can also be detected in the 77th frame, and they cannot be detected in the 110th frame. However, when they reappeared in the 190th frame, their IDs remained unchanged. The results show that the proposed method had good tracking performance compared to the original model.

#### 3.3.3. Evaluation on Pig Tracking Duration

In order to validate the tracker’s ability for continuous long-term tracking, this study conducted tracking tests on Video10 using both the StrongSORT and StrongSORT-Pig algorithms. The results are shown in Table 6.

For the 24 h long video tracking, it can be seen from Table 6 that the improved algorithm increases the HOTA by 41.2% and the IDF1 by 51.3%, and the MOTA and the MOTP are slightly improved. The IDSW is 0, indicating that there are no ID switches throughout the tracking process. Since no ID switching occurred during the whole process, it can be promised that the tracking result of the algorithm can serve as the basis for further analysis of the individual pigs.

#### 3.3.4. Comparison of Different Multi-Object Tracking Algorithms

To evaluate the performance of the proposed method in multi-pig tracking, we conducted tests on the multi-pig tracking dataset and compared with four state-of-the-art models in the field of multi-object tracking. The results are shown in Table 7.

According to Table 7, our proposed model StrongSORT-Pig demonstrates the best performance in terms of HOTA, IDF1, and IDSW metrics. Our proposed method increases the likelihood of successful matching. It achieves this by imposing restrictions on new trajectory generation and unmatched trajectory deletion. This ensures that the model does not endlessly add new trajectories and can only match among existing trajectories. Although our method does not achieve the best performance in terms of MOTA and MOTP metrics, it is only slightly lower than the highest scores. It is worth noting that C_BIOU also demonstrates good performance across various metrics, thanks to its utilization of buffer technology during the matching process. We can also consider incorporating this technique into the StrongSORT algorithm to further improve the effectiveness of multi-pig tracking.

To further validate the effectiveness of the proposed method in this study, we selected Video6 for visualization of the tracking results using BotSORT, DeepSORT, ByteTrack, C_BIOU, StrongSORT, and StrongSORT-Pig. The tracking results are shown in Figure 11.

We selected the tracking results in the 2555th frame and the 21,625th frame. From Figure 11, it can be observed that none of the six algorithms exhibited any instances of missed tracking. However, only our proposed algorithm achieved successful tracking, correctly matching all pigs. When tracking using the DeepSORT algorithm, all pigs failed to be tracked. With BotSORT and ByteTrack, only one pig was tracked successfully, and using the C_BIOU and StrongSORT algorithms, two pigs could be tracked successfully. In addition, as shown in Figure 11c, d, pigs numbered “5” and “6” exchanged IDs between the 2555th and 21,625th frames. This occurred because during a certain period of time, these two pigs were in close proximity, and both C_IOU and ByteTrack algorithms relied solely on motion models for matching without considering re-identification features to compute appearance similarity. In summary, compared to other multi-object tracking methods, our algorithm shows superior tracking performance in complex scenarios for pig tracking tasks, enabling the stable tracking of pigs.

### 3.4. Generalization

To validate the generalization of our research model, we conducted testing using unseen data from the training set. The test video had a duration of 30 min and was recorded in May 2022. The recording took place indoors with lighting provided by artificial sources. The occlusion in the video was not severe, and the camera was positioned at a 45-degree angle. It is important to note that the camera’s field of view did not cover the entire pig pen, which resulted in instances where pigs would momentarily disappear and then reappear in the frame.

#### 3.4.1. Generalization for Pig Detection

To validate the generalization ability of the pig detection model, we utilized ffmpeg to extract keyframes from the test videos. In order to maintain consistency with the original test set, a total of 360 images were selected for this testing. The model achieved Recall and mAP scores of 95.3% and 96.6%, respectively. The actual test results are shown in Figure 12.

From the provided figures, it can be observed that in Figure 12a, the occlusion is more severe compared to Figure 12b, and the detected confidence is lower. However, both images do not exhibit any cases of missed detections. Additionally, in comparison to the results in Section 3.2.2, Recall and mAP decreased by 3% and 1.8%, respectively. This could be attributed to the differences in the shooting angles between the training set and the test data used in this particular experiment.

In conclusion, based on the above analysis, the pig detection algorithm in this research demonstrates good generalization ability, as it successfully detects pigs even in the presence of occlusion and varying shooting angles between the training and test datasets.

#### 3.4.2. Generalization for Multi-Pig Tracking

We conducted testing on unseen videos from the training set using the StrongSORT-pig algorithm, and the results are shown in Figure 13. In the 34,625th frame, all pigs were successfully tracked. However, in the 35,775th frame, the pig with ID 3 disappeared. In the 36,925th frame, the previously disappeared pig reappeared but was not detected by the algorithm. Finally, in the 37,025th frame, the pig with ID 3 reappeared completely and was successfully detected, maintaining its original ID, resulting in successful tracking.

Compared to the results in Table 5, all performance metrics had decreased. HOTA, MOTA, IDF1, and MOTP had decreased by 2.8%, 3.9%, 2.9%, and 4.2%, respectively, reaching 79.8%, 93.7%, 88.2%, and 86.6%. The decrease in MOTP, which reflected the accuracy of the detection’s spatial position, is related to the pig detection model. The decrease in MOTA and IDF1 was due to the presence of a blind spot in the test video, where pigs disappear and cannot be detected, leading to changes in their IDs. The combined effect of these three metrics resulted in a decrease in HOTA.

Although the results obtained from testing in unseen scenarios were not as good as those from the seen scenarios, the StrongSORT-pig algorithm demonstrated its ability to effectively track pigs, even when they temporarily disappear and reappear in the video sequence.

## 4. Discussion

### 4.1. Discussion on Factors Affecting Pig Tracking Performance

In the multi-pig tracking task, there are many factors affecting the tracking effect, such as pig density, pig movement intensity, and the video frame rate.

It is worth noting that the nine videos, Video1–Video3, Video4–Video6, and Video7–Video9, were from three different scenes. In the scenes from Video 1 to Video 3, there are a large number of pigs present, with frequent occlusions, crowded environments, and active pig movements. In Video 7 to Video 9, the number of pigs is relatively small, and the scenes are relatively spacious. As for Video 4 to Video 6, these scenes have a moderate number of pigs, with a less crowded environment, and the level of pig movement is moderate. The results in Table 5 also indicate that the performance of the algorithm is influenced by the complexity of the scenes.

Particularly in the case of Video3, both HOTA and IDF1 are lower compared to the other videos. This is attributed not only to the pigs in this video exhibiting notably intense activity but also to its low frame rate of only 5 frames per second. The significant variations between adjacent frames make it challenging to accurately predict tracking trajectories, resulting in poorer algorithm performance.

In order to reduce the influence of these factors on the tracking performance, we can combine the detection model with dense optical flow, enhance the optical flow results by adding a mask to the detection frame, compute the motion of each detected object, and realize the prediction and matching of the target’s position between the front and back frames [29]. The introduction of optical flow can effectively alleviate the issue of large displacements of tracked objects in low-frame-rate videos.

### 4.2. Discussion on the Application of Long-Term Pig Tracking Algorithm

Although it has been difficult for existing research to achieve the long-term regular analysis of individual pigs [30], it is valuable for behavioral studies and is beneficial for a deeper understanding of the health status of pigs [31]. In this study, we realized the all-day tracking of pigs. Based on this, we can use the results for analyzing the behavioral rhythms of individual pigs throughout the day, as a basis for detecting pig abnormalities, and early health warnings.

The pen was divided into four sections based on the positions of different areas in the video frames, as shown in Figure 14. The area of (490,200)~(720,975) was classified as the defecation area, (720,200)~(1480,975) was classified as the resting area, (1480,200)~(1580,975) was classified as the eating area, and the remaining parts of the video frames that were unrelated to this experiment were considered irrelevant zones. To further validate the practicality of the tracking algorithm, this study counted the time spent by each pig in different areas (resting area, eating area, and defecating area) based on the 24 h tracking of the pigs. The results were compared with the observed results, which manually recorded the time spent by each pig in each area, and the duration of each area was added up to calculate the total time spent.

The data were preprocessed according to the following procedure: (a) Grouping the tracking results by pig ID (1, 2, 3, 4). (b) Calculating the center-point positions of the bounding boxes for each pig. (c) Filling in missing data. The pig detector is not fully able to detect pigs in all scenarios, and there are very few frames in which a particular pig cannot be detected during the tracking process, so filling in the missing frames ensures that there are detection data for every frame of every pig to facilitate subsequent grouping based on time. (d) Aggregating the data for each pig every 1 s.

After preprocessing the tracking results, the area where the pig is located every second can be determined by the position of the center point of the pig bounding box. The specific steps were as follows: (a) Determine frame by frame the area where the center point of the pig detection frame is located. (b) Identify the area with the highest proportion within that second (25 frames) to determine the pig’s location for that second. (c) Summarize the time each pig stays in each area.

It should be noted that due to the distance between the center point of the detection box and the pig’s head, it may not accurately reflect the pig’s actual position when the pig is eating. Therefore, adjustments to the eating area will be needed. In the case of pigs eating, the length of the pig’s body was measured, and the average distance from the center point of the detection box to the head was 230 pixels. So, the left boundary of the eating area was extended outward to a position of 1350 pixels as the criterion for determining that the pig is in the feeding area.

From Table 8, it can be seen that all pigs spent the most time in the resting area throughout the entire day, which is consistent with their natural behavior. In addition, there are differences in the amount of time each pig spent in different areas, which might be related to individual differences. The more intuitive comparison of the statistical results of the two methods is shown in Figure 15. The four bar charts represent the comparison of the time each of the four pigs spent in different areas. It is evident from the charts that our statistical results differ very little from the manually observed results.

Table 9 shows the error between the video tracking method and the manual observation method for each pig in each area. The table indicates that there are slight differences between the two methods, with an overall error of 1.55%, which falls within an acceptable range. For all pigs, pig 4 has the largest statistical error, but it is only 2.44%. For each area, the statistical error is highest in the eating area, at 2.70%, while the other two areas are within 2%, demonstrating good consistency. Overall, this set of data show that both methods have very small errors, validating the accuracy and reliability of the results. This also indicates that pig tracking has certain application value in the field of pig farming.

In terms of pig tracking algorithm applications, Huang et al. [32] combined YOLO v5 with DeepSORT to achieve efficient pig counting through tracking, but they have not incorporated behavior analysis of the pigs. Additionally, Tu et al. [33] combined behavior recognition with pig tracking, using a pig detection model to extract pig behaviors and a pig tracking algorithm to determine pig identities, enabling static behavior tracking of the pigs. Zhang et al. [34] combined pig tracking with the Slowfast action recognition network, achieving the tracking of complex pig behaviors. However, due to overlapping and occlusion among the pigs, it is challenging to achieve long-term tracking. By improving the StrongSORT algorithm, our study enables accurate tracking for 24 h, laying the foundation for long-term pig tracking applications.

## 5. Conclusions

This paper proposed a multi-pig tracking algorithm StrongSORT-Pig and explored its application in pig farming. The core technical contributions are as follows:We proposed a lightweight pig detection model, YOLO v7-tiny_Pig, which reduced the number of parameters by 36.7% and improved the recognition speed by 17.6% while ensuring detection accuracy.The StrongSORT multi-object tracking algorithm was optimized to suit the characteristics of pig tracking tasks. After testing on datasets from different scenarios, the optimized pig tracking algorithm achieved HOTA and IDF1 scores of 83.16% and 91.42%, respectively, which improved by 6.19% and 10.89%. The IDSW was reduced by 69%, significantly enhancing the algorithm’s stability and enabling the long-term tracking of pigs.This study enabled the continuous video tracking of pigs throughout the day. Our automated tracking method calculated pigs’ stay times in different areas of the pigsty and achieved an overall error rate of 1.55% compared to manual counting, demonstrating a high degree of correlation. The conclusions drawn from the automated tracking results also aligned with pig behavior patterns.

In summary, the multi-pig tracking method proposed in this study demonstrates high practicality, catering to the needs of practical production. It improves the efficiency of farming personnel and lays the foundation for the automated monitoring of pigs.

## Figures and Tables

**Figure 1 animals-14-01505-f001:**
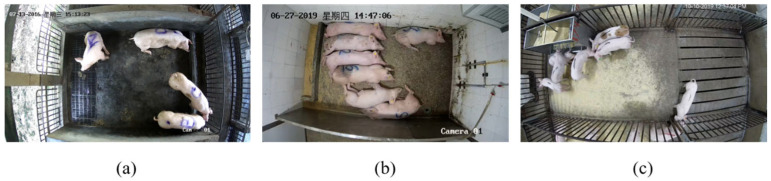
View of each pig farm: (**a**) Luogang Pig Farm, (**b**) Xinxing Pig Farm, (**c**) the public videos.

**Figure 2 animals-14-01505-f002:**
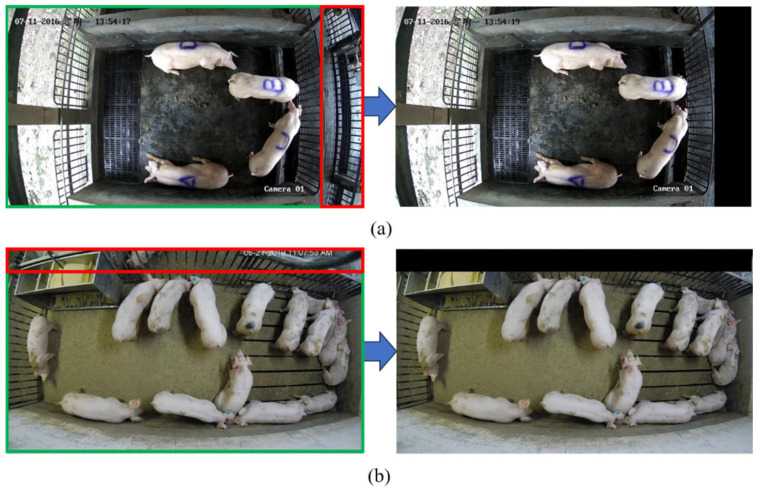
Comparison of the effect before and after preprocessing: (**a**) Luogang Pig Farm, (**b**) public videos.

**Figure 3 animals-14-01505-f003:**
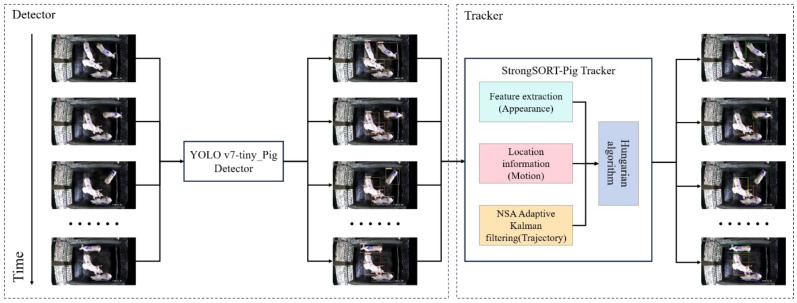
The framework of multi-pig tracking algorithm.

**Figure 4 animals-14-01505-f004:**
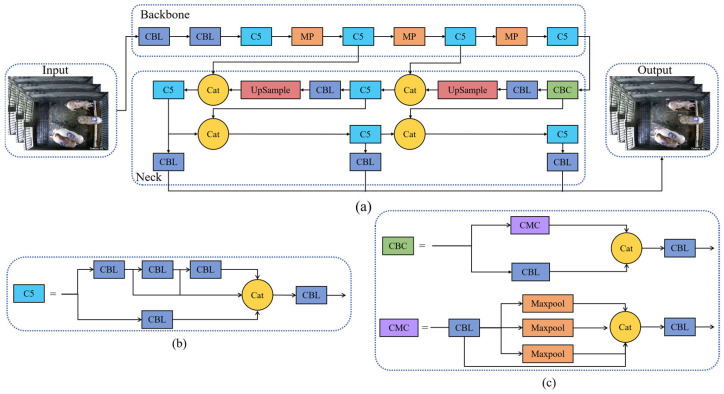
YOLO v7-tiny network architecture. (**a**) YOLO v7-tiny model structure, (**b**) C5 Network structure, (**c**) CBC Network structure.

**Figure 5 animals-14-01505-f005:**
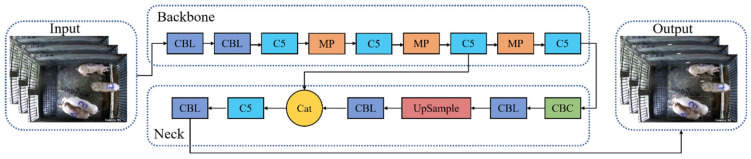
The network structure of YOLO v7-tiny_Pig.

**Figure 6 animals-14-01505-f006:**
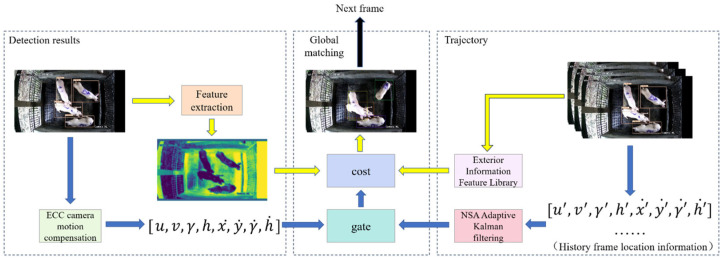
Flow chart of multi-pig tracking process.

**Figure 7 animals-14-01505-f007:**
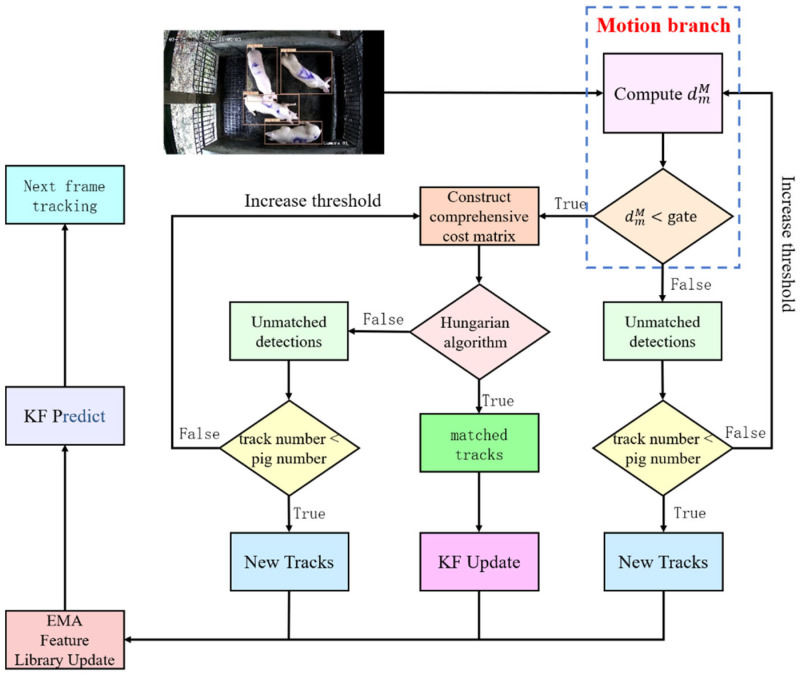
Global matching flowchart.

**Figure 8 animals-14-01505-f008:**
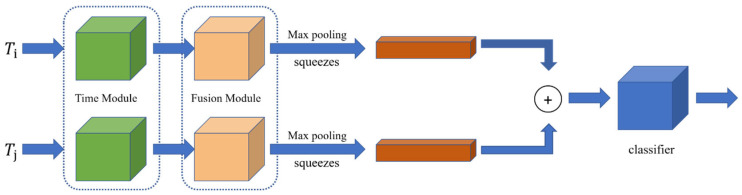
AFLink module.

**Figure 9 animals-14-01505-f009:**
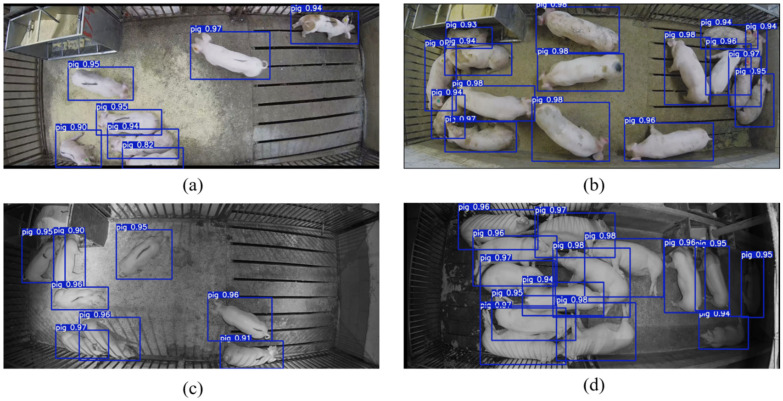
Example of pig detection results based on YOLO v7-tiny_Pig. (**a**) Sparse during the day, (**b**) dense during the day, (**c**) sparse at night, (**d**) dense at night.

**Figure 10 animals-14-01505-f010:**
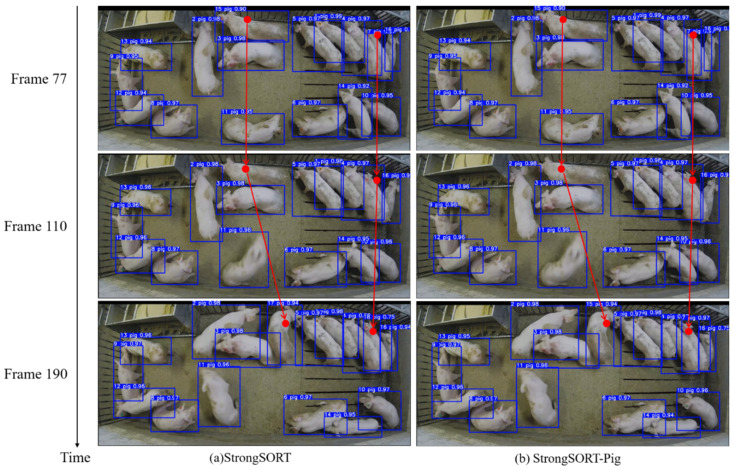
The tracking results of the proposed model compared to the original model.

**Figure 11 animals-14-01505-f011:**
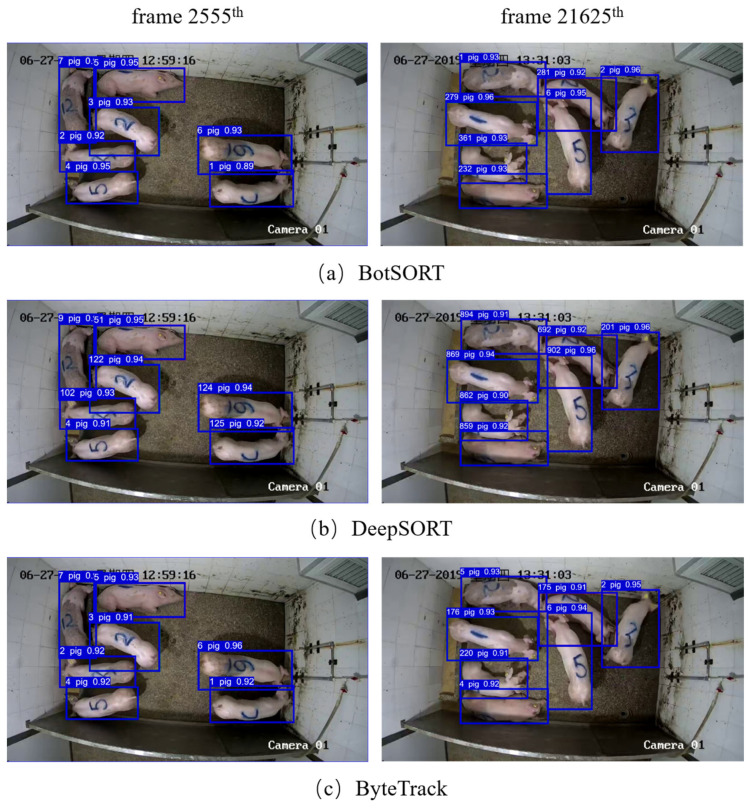
Comparative example of tracking results for different tracking algorithms.

**Figure 12 animals-14-01505-f012:**
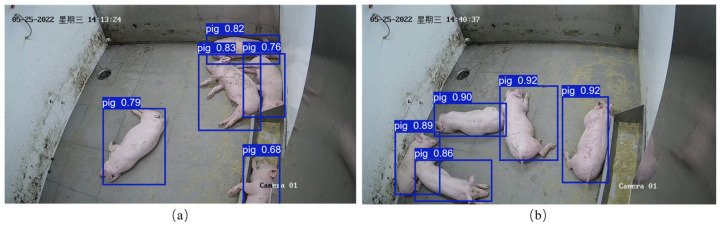
Generalization experiment results for pig detection. (**a**) Dense pig scene, (**b**) Sparse pig scene.

**Figure 13 animals-14-01505-f013:**
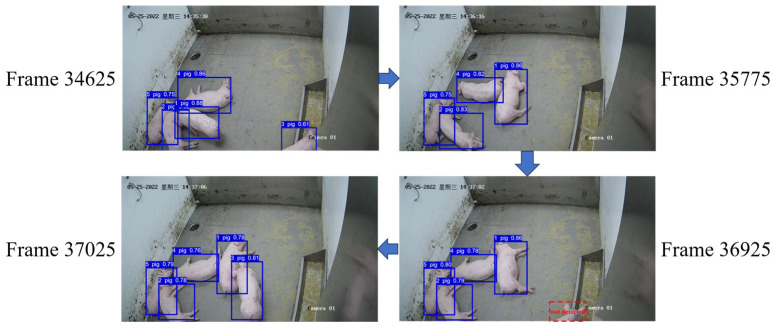
Generalization experiment results for multi-pig tracking.

**Figure 14 animals-14-01505-f014:**
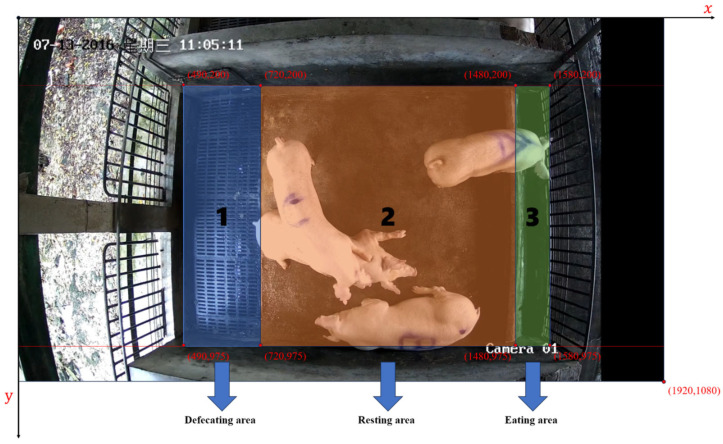
Area division of pig pens.

**Figure 15 animals-14-01505-f015:**
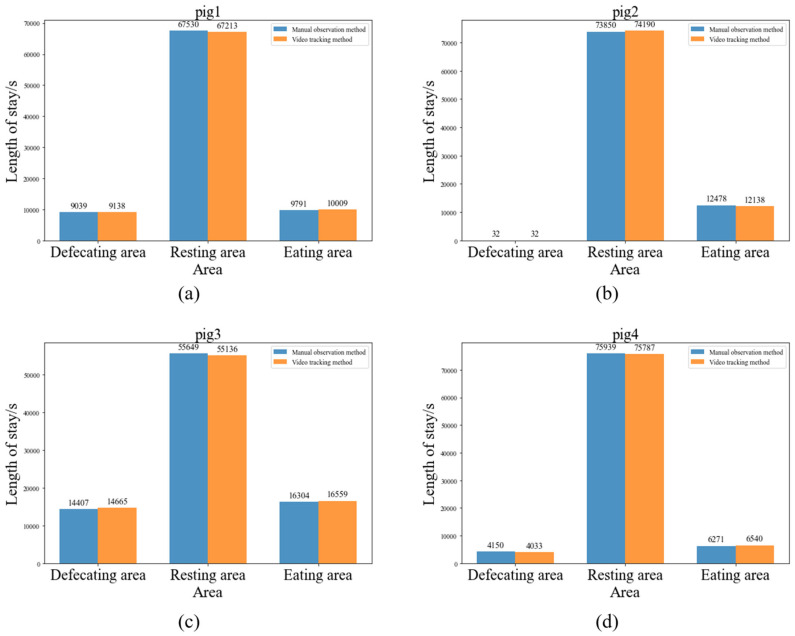
Comparison of length of stay. (**a**) pig1, (**b**) pig2, (**c**)pig3, (**d**) pig4.

**Table 1 animals-14-01505-t001:** Information on each video collection.

Video Collection	Experimental Pig Farm	Number of Pigs	Light Conditions	Pig Pen Size
Self-collected videos	Luogang Pig Farm	4	Daytime and nighttime	4 m×5 m
Self-collected videos	Xinxing Pig Farm	7	Daytime	2 m×4 m
Public videos	Not mentioned	7–16	Daytime and nighttime	Not mentioned

**Table 2 animals-14-01505-t002:** The pig detection datasets.

Dataset	Experimental Pig Farm	Number of Pictures	Total
Training	Luogang Pig Farm	1440	4320
Training	Xinxing Pig Farm	1440	
Training	The public videos	1440	
Testing	Luogang Pig Farm	360	1080
Testing	Xinxing Pig Farm	360	
Testing	The public videos	360	

**Table 3 animals-14-01505-t003:** The pig tracking datasets.

Video No.	Duration	Frame Rate	Light Conditions	Density Level	Experimental Pig Farm
1	2 min	5	daytime	dense	The public videos
2	10 min	5	nighttime	dense	The public videos
3	30 min	5	daytime	dense	The public videos
4	2 min	10	daytime	dense	Xinxing Pig Farm
5	10 min	10	daytime	dense	Xinxing Pig Farm
6	1 h	10	daytime	dense	Xinxing Pig Farm
7	2 min	25	daytime	sparse	Luogang Pig Farm
8	10 min	25	nighttime	sparse	Luogang Pig Farm
9	1 h	25	nighttime	sparse	Luogang Pig Farm
10	1 day (24 h)	25	daytime and nighttime	sparse	Luogang Pig Farm

**Table 4 animals-14-01505-t004:** Performance comparison of different pig detection models. P: Precision; R: Recall; F1: F1 Score; FPS: frames per second; mAP: Mean Average Precision.

Models	P (%)	R (%)	F1 (%)	$\mathrm{FPS}\;(f\cdot s^{-1}$)	mAP (%)	Parameter
YOLO v7	99.7	97.9	98.8	127	99.0	$3.7\times 10^{7}$
YOLO v7-tiny	99.5	98.4	98.9	370	98.6	$6\times 10^{6}$
YOLO v7-tiny_Pig	99.3	98.3	98.8	435	98.4	$3.8\times 10^{6}$

**Table 5 animals-14-01505-t005:** Performance comparison of different multi-pig tracking models. HOTA: Higher-Order Tracking Accuracy; MOTA: Multiple Object Tracking Accuracy; IDF1: Identification F1; MOTP: Multiple Object Tracking Precision; IDSW: Identity Switch.

Algorithms	Video No.	HOTA (%)	MOTA (%)	IDF1 (%)	MOTP (%)	IDSW (Times)	FPS $(f\cdot s^{-1}$)
StrongSORT	1	72.2	91.7	79.0	88.6	25	26.3
	2	68.4	96.2	72.2	86.0	41	24.6
	3	49.2	98.9	42.6	93.7	47	22.5
	4	89.7	98.9	97.9	91.3	3	25.8
	5	85.8	98.0	92.8	90.4	14	25.2
	6	67.5	98.7	62.7	92.8	23	23.6
	7	89.4	99.6	87.6	94.3	1	29.6
	8	90.8	99.1	99.5	91.2	0	28.7
	9	79.7	99.1	90.5	89.4	17	27.8
	Average/Total	76.97	97.8	80.53	90.86	171	26.01
StrongSORT-Pig	1	77.8	90.5	89.7	88.7	14	28.6
	2	73.3	95.3	84.5	86.0	43	27.3
	3	62.9	98.7	64.2	93.8	22	26.8
	4	90.8	98.9	99.4	91.3	2	27.2
	5	89.1	98.0	99.0	90.4	8	26.9
	6	83.2	98.7	91.2	92.8	12	26.5
	7	95.1	99.6	99.8	94.3	0	32.6
	8	91.0	99.6	99.8	91.2	0	32.4
	9	80.9	99.2	92.3	89.4	2	30.9
	Average/Total	82.68	97.6	91.1	90.88	103	28.8
StrongSORT-Pig_Plus	1	78.9	91.7	90.2	88.6	8	23.2
	2	73.9	96.5	85.2	85.8	18	20.8
	3	62.8	98.9	64.3	93.4	20	18.9
	4	90.8	99.2	99.6	91.0	0	22.7
	5	89.9	98.2	99.1	90.4	0	21.8
	6	83.1	99.0	91.4	92.4	7	19.2
	7	95.2	100.0	100.0	94.1	0	25.2
	8	91.5	99.9	99.9	91.3	0	24.9
	9	82.3	99.3	93.1	89.3	0	22.5
	Average/Total	83.16	98.1	91.42	90.7	53	22.13

**Table 6 animals-14-01505-t006:** Performance comparison of long-term tracking results of different algorithms. HOTA: Higher-Order Tracking Accuracy; MOTA: Multiple Object Tracking Accuracy; IDF1: Identification F1; MOTP: Multiple Object Tracking Precision; IDSW: Identity Switch.

Algorithms	HOTA (%)	MOTA (%)	IDF1 (%)	MOTP (%)	IDSW (Times)	FPS $(f\cdot s^{-1}$)
StrongSORT	56.3	97.4	47.6	98.9	33	25.6
StrongSORT-Pig	97.5	97.6	98.9	99	0	28.1

**Table 7 animals-14-01505-t007:** Tracking results of different algorithms for multi-pig tracking. HOTA: Higher-Order Tracking Accuracy; MOTA: Multiple Object Tracking Accuracy; IDF1: Identification F1; MOTP: Multiple Object Tracking Precision; IDSW: Identity Switch.

Algorithms	HOTA (%)	MOTA (%)	IDF1 (%)	MOTP (%)	IDSW (Times)	FPS $\left(f\cdot s^{-1}\right)$
BotSORT	75.83	94.8	82.94	91.3	155	23.73
DeepSORT	71.68	88.63	78.63	90.62	252	28.68
ByteTrack	73.3	91.82	84.82	89.2	128	32.21
C_BIOU	75.26	95.38	85.6	92.12	138	33.85
StrongSORT	76.97	97.8	80.53	90.86	171	26.01
StrongSORT-Pig	82.68	97.6	91.1	90.88	103	28.8

**Table 8 animals-14-01505-t008:** Duration of stay statistics in each area.

Methods	Pig Identity	Defecating Area Time (s)	Resting Area Time (s)	Eating Area Time (s)
Manual observation method	1	9039	67,530	9791
	2	32	73,850	12,478
	3	14,407	55,649	16,304
	4	4150	75,939	6271
Video tracking method	1	9138	67,213	10,009
	2	32	74,190	12,138
	3	14,665	55,136	16,559
	4	4033	75,787	6540

**Table 9 animals-14-01505-t009:** Error analysis of the multi-pig tracking method.

Pig Identity	Error in Defecating Area (%)	Error in Resting Area Time (%)	Error in Eating Area Time (%)	Average Error (%)
1	1.10	0.47	2.22	1.26
2	0	0.46	2.72	1.06
3	1.79	0.92	1.56	1.42
4	2.82	0.20	4.29	2.44
Average	1.43	0.51	2.70	1.55

## Data Availability

Data are contained within the article.

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
