# Peer review of "A Long-Term Video Tracking Method for Group-Housed Pigs"

_animals, 2024, doi:10.3390/ani14101505_

Round 1

Reviewer 1 Report

Comments and Suggestions for Authors

1L361-L363Regarding evaluation metrics for object detection, Precision, Recall, F1 Score, and FPS are mentioned, but the calculation formula for FPS is not listed, and the commonly used mAP metric in object detection is also not mentioned. Please add the formulas for FPS and mAP, and complete the data for the mAP metric in Table 4.

2L429-L435 describes the correction algorithm, StrongSORT-Pig_Plus, and Table 5 also presents corresponding data. It shows better tracking performance compared to the StrongSORT-Pig algorithm. However, there is no mention of why the StrongSORT-Pig_Plus algorithm is not used in this paper. Please provide specific reasons for not using the StrongSORT-Pig_Plus algorithm.

3L489-L503 describes the issue of ID switches for different multi-object tracking algorithms at specified frames. However, the numbering of subfigures a, b, c, and d in Figure 11 is too blurry to read the corresponding information from the image. Please replace the image with a clearer one.

Reviewer 2 Report

Comments and Suggestions for Authors

This study introduces an enhanced tracking method tailored for long-term monitoring of group-housed pigs, tackling common obstacles like occlusion and motion blur in farm settings.The enhanced system delivers notable improvements in tracking metrics, with High Order Tracking Accuracy (HOTA), Multi-Object Tracking Accuracy (MOTP), and Identification F1 (IDF1) scores at 83.16%, 97.6%, and 91.42%, respectively. In addition to presenting this method, the authors have made their 24-hour pig tracking video dataset publicly available, contributing to the precision livestock farming research community. However, before publication, there are suggestions for improvement.

In line 45-49, please add more details for current research.

Please mention video collection time in section 2.1

please mention how you collect public video dataset in 2.2.2

in 3.2.1 please add content how you calculate FPS

in your 4.2 discussion, please at least compare your work with 3 previous studies

Reviewer 3 Report

Comments and Suggestions for Authors

This manuscript improved the accuracy of "seen(i.e., train with Chinese/German pigs -> test with Chinese/German pigs)" scenarios.

For practical applications, however, authors need to improve the accuracy of "unseen(i.e., train with Chinese pigs -> test with German pigs, vice versa)" scenarios. At least, authors need to show the accuracy of these unseen scenarios in Experimental Result section.
